# Metabolomic Studies in Inborn Errors of Metabolism: Last Years and Future Perspectives

**DOI:** 10.3390/metabo13030447

**Published:** 2023-03-18

**Authors:** Marcello Cossu, Roberta Pintus, Marco Zaffanello, Michele Mussap, Fabiola Serra, Maria Antonietta Marcialis, Vassilios Fanos

**Affiliations:** 1School of Pediatrics, University of Cagliari, 09042 Monserrato, Italy; 2Department of Surgical Science, University of Cagliari, 09042 Monserrato, Italy; 3Department of Surgical Science, Dentistry, Gynecology and Pediatrics, University of Verona, 37100 Verona, Italy; 4Laboratory Medicine, Department of Surgical Sciences, University of Cagliari, 09042 Monserrato, Italy; 5Neonatal Intensive Care Unit, AOU Cagliari, 09042 Monserrato, Italy; 6Neonatal Intensive Care Unit, Department of Surgery, University of Cagliari, 09042 Monserrato, Italy

**Keywords:** inborn errors of metabolism, metabolomic, newborn screening, biomarkers

## Abstract

The inborn errors of metabolism (IEMs or Inherited Metabolic Disorders) are a heterogeneous group of diseases caused by a deficit of some specific metabolic pathways. IEMs may present with multiple overlapping symptoms, sometimes difficult delayed diagnosis and postponed therapies. Additionally, many IEMs are not covered in newborn screening and the diagnostic profiling in the metabolic laboratory is indispensable to reach a correct diagnosis. In recent years, Metabolomics helped to obtain a better understanding of pathogenesis and pathophysiology of IEMs, by validating diagnostic biomarkers, discovering new specific metabolic patterns and new IEMs itself. The expansion of Metabolomics in clinical biochemistry and laboratory medicine has brought these approaches in clinical practice as part of newborn screenings, as an exam for differential diagnosis between IEMs, and evaluation of metabolites in follow up as markers of severity or therapies efficacy. Lastly, several research groups are trying to profile metabolomics data in platforms to have a holistic vision of the metabolic, proteomic and genomic pathways of every single patient. In 2018 this team has made a review of literature to understand the value of Metabolomics in IEMs. Our review offers an update on use and perspectives of metabolomics in IEMs, with an overview of the studies available from 2018 to 2022.

## 1. Introduction

The Inborn errors of metabolism (IEMs or Inherited Metabolic Disorders) are a heterogeneous group of genetic disorders, usually inherited as recessive disorders or, in smaller proportion, as the result of spontaneous mutations. These diseases are caused by a defect in metabolic pathways involved in the metabolism of small (i.e., amino acids, fatty acids, etc.) and large molecules (i.e., glycogen, sphingolipids, etc.), coenzymes, vitamins, etc. The defect results in an alteration of the body concentrations (increase or reduction) of these molecules. This may lead, for instance, to an excessive accumulation of macromolecules, or a deficit of the energy pathways, etc. The clinical expression of the defect and the organ damage depend on the metabolic pathway involved [1,2].

Although IEMs are disorders genetically transmitted the model ‘one gene-one enzyme-one disease’ can no longer be considered appropriate [3].

Indeed, each patient has a peculiar phenotype that requires a personalized approach to these complex and rare diseases. Thus, Metabolomics, the global study of metabolites, small molecules within a mass range of 50 or 1500 daltons, can plays a central role in the study of IEMs, leading to a better understanding of the pathogenesis and pathophysiology, contributing to improving the processes of diagnosis, therapy and prognosis. Integrating genomics and metabolomics data represents the current challenge to improve the diagnosis and prognosis of IEM [3,4,5].

The purpose of this narrative review is to offer an update on the use and perspectives of metabolomics at IEMs.

This review will focus on an overview of comprehensive studies, available from 2018 to 2022, and some interesting studies from 2017.

### 1.1. Inborn Errors of Metabolism

IEMs are monogenic, often untreatable, disorders, in which a single enzyme or protein causes a distinct metabolic signature. Different variants in some proteins can lead to either deficiency or increase activity, with drastically different phenotypes. Due to their remarkable heterogeneity, IEMs are classified according to clinical, pathophysiological, affected enzymes, affected organelles and many other criteria. Nevertheless, numerous biochemical phenotypes that do not cause clinical disease have been described [6].

Saudubray, et al. classified IEMs into three groups: intermediate metabolism disorders affecting small molecules, disorders involving primarily energy metabolism, and disorders involving complex molecules [7]. Furthermore, Ferreira et al. have proposed a new “International classification of Inherited metabolic disorders (ICIMD)”, that includes 1450 disorders divided in 24 categories comprising 124 groups. They reported IMDs that involved several metabolic pathways including neurotransmitter and endocrine metabolism and metabolic cell signaling [6].

Taken alone, IEMs are rare, however, globally they affect about 0.5–1 in 1000 people [5,8].

For these reasons, awareness of these diseases, their presentations, and their evaluation is critical for healthcare professionals because they can occur at any age [1].

The investigation of Inborn Errors of Metabolism has already benefitted from metabolomics as a tool to facilitate diagnosis by finding appropriate biomarkers in human samples [9].

### 1.2. Metabolomics

Metabolomics, one of the “omics” sciences, is a technique that investigates and detects the whole and comprehensive set of metabolites (including sugars, lipids, small peptides, vitamins and amino acids) present in a given biological system, organs, and biological fluids, portraying their phenotypes. This has proven to be very useful in the investigation of the human health since this technique is able to spot very rapidly every metabolic change induced by pathophysiological stimuli such as diseases, drug administration, environmental changes and nutrition [10].

The study of the complex metabolic fingerprint and the comparative analyses of metabolomes are performed by a combined approach of spectrometric and spectroscopic techniques and computer programs. The most common metabolomics analyses techniques are nuclear magnetic resonance (NMR) spectroscopy, gas or liquid chromatography-mass spectrometry (GC-MS and LC-MS) [10,11].

Both NMR spectroscopy and mass spectrometry (MS) are widely used in metabolomics research. NMR spectroscopy is a powerful technique for the identification of small molecules and the analysis of their structure and dynamics. It is a non-destructive technique that does not require the separation of the metabolites prior to analysis, making it a suitable tool for the analysis of complex mixtures, such as biological fluids. In contrast, MS is a destructive technique that requires the separation of the metabolites before analysis. The separation step is usually performed using either gas or liquid chromatography. MS is capable of providing more accurate quantification of metabolites, and can detect a wider range of metabolites than NMR spectroscopy.

According to Emwas et al., NMR spectroscopy is best suited for the analysis of low molecular weight compounds, while MS is more suitable for the analysis of high molecular weight compounds. NMR spectroscopy is also less sensitive than MS, which can be a limitation when analyzing low-abundance metabolites. However, NMR spectroscopy is more robust and reproducible than MS, making it a valuable tool for large-scale metabolomics studies.

In summary, both NMR spectroscopy and MS have their strengths and weaknesses, and the choice of which technique to use depends on the specific goals of the study and the characteristics of the metabolites being analyzed [12,13,14].

## 2. Inborn Error of Metabolism and Metabolomic

Timely newborn screening and genetic profiling are crucial in early recognition and treatment of IEMs. Diagnostic delay may result in acute metabolic decompensation, progressive neurologic injury, or death [10].

In most developed countries, and many developing countries, each newborn is screened for IEMs by dry blood spot (DBS) testing. In 1980s, targeted metabolomics with tandem mass spectrometry (MS/MS) was introduced as a reference technique for newborn screening, because of its capacity to screen a wide range of previously unscreened IEMs by a single test performed on a DBS.

From their introduction in public health care, newborn screening programs have expanded to include now more than 50 inherited metabolic disorders, including aminoacidopathies, organic acidemias, fatty acid oxidation disorders, and lysosomal storage disorders. Despite its absolute clinical value and its primary role in public health care, newborn screening offers only a snapshot of minimal subset of metabolic alterations, and several IEMs are not identifiable by routine newborn screening programs [5].

The laboratory approach for IEMs screening can be schematically represented by four steps: (I) general metabolic screening test; (II) specific metabolite assays; (III) enzymes studies; (IV) DNA analysis [5].

Currently, MS/MS techniques are routinely used for neonatal screening of IEMs, since they are rapid, susceptible, and specific, requiring shallow sample volume and offering a high throughput [5].

Moreover, the exclusive use of targeted metabolomic assays can focus only on limited panels of compounds for selected IEM diseases. Such targeted approaches face the problem of false negative and false positive diagnoses that could be overcome if metabolic screening adopted analyses of a broader range of analytics.

In current practice, clinical symptomatology determines the selection of specific analyses for an individual patient. However, this strategy heavily relies on the completeness of clinical information provided to the metabolic laboratory and therefore holds the risk of false-negatives, if a metabolic test has not been performed due to an incomplete description of patient symptoms. For the IEMs that are not included in newborn screening, diagnostic profiling in the metabolic laboratory is indispensable to reach a correct diagnosis for an individual patient.

Additionally, IEMs may present with multiple overlapping symptoms and metabolites and the discovery of novel metabolic defects is likely hampered by investigating only a limited selection of known metabolic pathways [2,15].

The use of untargeted metabolomics could implement current targeted metabolite assays.

Untargeted metabolomics do not rely on predefined target lists and can detect as many metabolites as possible in a sample, allowing to screen for many metabolic pathways simultaneously [2].

Untargeted metabolomics methods can be merged with aspects of classic targeted assays by validating methods for specific known IEM biomarkers. Furthermore, it can highlight pathways of disease and support new discoveries that may lead to new treatment options [16].

## 3. Targeted Metabolomics

In the diagnostic process, metabolomics plays a role, besides the screenings, also in the detection of new specific markers.

For example, in organic acidemias and aminoacidopathies, it helped to detect the biomarker profile for the holo-carboxylase synthase deficiency spectrum. In other IEMs, they also see that changes in metabolite profiles differentiated post-treatments and follow-ups while allowing for the establishment of different clinical states of a biochemical disorder [17].

Other IEMs difficult to manage, due to lack of understanding of their biochemical pathogenesis and the absence of severity markers, are Mitochondrial disorders.

In a proteomics study on the mitochondrial encephalomyopathy lactic acidosis and stroke-like episodes (MELAS) and targeted and untargeted metabolomics, they were able to validate 20 analytes (1 protein, 19 metabolites) on plasma. Many of these 20 analytes correlate strongly with established measures of severity, and provide evidence of NADH redox imbalance as a predominant feature of mitochondrial respiratory chain disease. These biochemical insights, combined with the ability to track disease severity, should be helpful in the development of new treatment strategies for mitochondrial disorders [18].

Targeted metabolomics can also help in the evaluation of therapy adherence. For example, it was used in a study on the pharmacological and dietary adherence in patients with hereditary tyrosinemia type 1 [19].

### 3.1. Metabolomics Studies with Cerebrospinal Fluid Samples

In IEMs affecting neurological features, the sample for laboratory investigations could be cerebrospinal fluid (CSF). Nevertheless, to date, available tests are currently performed on different analytical platforms, requiring a large sample volume and long turnaround time, which often delays timely diagnosis (Table 1) [20].

However, Haijes et al., in 2019, showed that the direct-infusion high-resolution mass spectrometry (DI-HRMS) based non-quantitative metabolomics could correctly capture the biochemical profile of patients with an IEM in CSF. Eleven patient samples, harboring eight different IEM, and thirty control samples were analyzed. This analysis of patient CSF profiles resulted in correct ‘most probable diagnoses’ for all eleven patients, including non-ketotic hyperglycinaemia, propionic aciduria, purine nucleoside phosphorylase deficiency, argininosuccinic aciduria, tyrosinaemia type I, hyperphenylalaninemia and hypermethioninaemia [21].

Therefore, Klinke et al., in 2020, developed an extendable targeted LC-MS/MS platform for the analysis of multiple metabolites in CSF, providing reference concentrations for several biomarkers. They demonstrated that a smaller sample volume (200 μL) is required with a single analytical platform resulting in faster diagnostic results while reducing preanalytical problems [20].

Then, the cerebrospinal fluid could be a very useful sample in those IEMs where there’s still a lack of diagnostic test or those whom the differential diagnosis is difficult.

### 3.2. The Lyso-Gb3 Case in Fabry Disease

Even in a well-known IEMs like Fabry disease (AFD), the role of metabolomics is still relevant. For example, in studies of complications of AFD, including major renal, cardiac and cerebrovascular events, the progressive accumulation of a globotriaosylceramide (Gb3) in numerous cell types and of deacylated Gb3 (globotriaosylsphingosine, lyso-Gb3) in the circulation, was studied. Lyso-Gb3 was evaluated for a better understanding of the FD, in the follow-up of these patients and for the evaluation of the therapeutic efficacy and their changes.

Indeed, Christiane Auray-Blais et al., using a targeted metabolomic approach, found that Lyso-Gb3 is positively associated with LVMI (left ventricular mass index) and a clinical scoring system (MMSSI) in AFD adults carrying the IVS4 late-onset cardiac variant mutation (Table 2) [22,23,24].

Moreover, metabolomics can also help in the evaluation of the current treatment options for FD, which include two different recombinant enzyme replacement therapies (ERTs) (agalsidase-α and agalsidase-β), as well as chaperone therapy (migalastat). The ERTs with agalsidase-β is associated with a significantly lower incidence of renal, cardiovascular, and cerebrovascular events in patients. In these cases, Lyso-Gb3 has been suggested as a possible marker of efficacy of ERT treatment because the plasma levels were considerably reduced, when the therapy was effective in reducing the organs’ injury [22,25].
metabolites-13-00447-t002_Table 2Table 2Relevant targeted metabolomics studies in Fabry disease investigation.Author and YearPatientsSampleMethodsRelevant MetabolitesBiological MeaningHitoshi Sakuraba et al., 2018[23]161: 100 controls and 61 patients with FDPlasmaLC–MS/MS↑Lyso-Gb3 in:-FD patient than controls: higher in the classic one than in the late-onset.-Fabry females who developed renal involvement was statistically higher than in those without this defect-in all the cases that continuously developed anti-agalsidase alfa antibodies↓Lyso-Gb3: in all FD treated with ERT, lower in the patients with anti-agalsidase alfa antibodiesMarker of disesease, renal disfunction and positivity to anti-agalisidase alfa antibodies.Marker in the ERT therapy.Christiane Auray-Blais et al., 2017[24]191 patients with FD carrying the IVS4 late onset cardiac variant mutation.Urine and plasma prior ERT initiationMS (GB3)UPLC-MS/MS (Lyso-Gb3)↑Plasma lyso-Gb3;↑Urine lyso-Gb3 at m/z (+16), (+34), and (+50).Positively associated with LVMI (left ventricular mass index) and MSSI (clinical scoring system) in adults, mainly concerning disease severity and predictive valueDerralynn A Hughes et al., 2017[26]57 (36 patients switched from ERT to migalastat and 21 remained on ERT)PlasmaLC-Lyso-Gb3 in coorts: same level, low and stableUseful for the monitoring and changes of therapy in FDHiroki Maruyama et al., 2019[27]2360PlasmaUPLC-MS/MS↑lyso-Gb3 levels in 23 in patients with FD.Promising FD primary screening biomarker↑: increased value, ↓: decresed value.


Nevertheless, a possible complication of ERT is the production of antibodies. In this condition, the plasma level of lyso-Gb3 increased again and remained high. With the acquisition of immunotolerance, the plasma levels of the marker decreased again with a gradual decrease and stabilization of the plasma level of lyso-Gb3, respectively, after ERT [23].

In another study, authors tried to use the Lyso-Gb3 for the evaluation of the therapy modification from ERT (a lifelong biweekly intravenous infusion administration), to migalastat, an oral pharmacological chaperone. In patients with amenable mutations, the Migalastat and ERT had similar effects on Fabry disease substrate, and the plasma lyso-Gb3 stayed at the same low levels as ERT [26]. Moreover, in patients with non-amenable mutations, plasma lyso-Gb3 increased when they switched from ERT to Migalastat: this difference in the two patients’ cohorts can support the theory of the clinical validity of plasma Lyso-Gb3 as a severity and therapy marker.

A recent study examined its applicability as a primary screening biomarker for classic and late-onset Fabry disease in males and females, with the measure of lyso-Gb3 by ultra-performance liquid chromatography/tandem mass spectrometry [26]. The authors showed that males with classic FD showed higher levels of lyso-Gb3 than those with late-onset FD. However, they did not detect any significant difference of Lyso-Gb3 in female probands [27].

### 3.3. Metabolomics Investigations to Monitor the Effect of Different Nutritional Regimens in IEMs Patients

In many IEMs, a primary role is played by specific diets in order to avoid or increase some specific element.

For example, phenylketonuria is an inherited disorder characterized by a loss of function of hepatic phenylalanine hydroxylase in which the initiation of a low-phenylalanine (Phe) diet in infancy is required to prevent severe cognitive deterioration caused by accumulation of Phe metabolites in the brain (Table 3) [28,29].

A strict low-Phe diet eliminates all animal proteins and consists of a controlled amount of natural protein from plant sources to provide minimum Phe requirements and relies on the consumption of either elemental amino acid medical foods (AA-MFs) or glycomacropeptide medical foods (GMP-MFs) to provide the majority of dietary nitrogen and variable amounts of calories, carbohydrates, fats, minerals, and micronutrients [29,30].
metabolites-13-00447-t003_Table 3Table 3Relevant metabolomics studies applied to different nutritional regimens in IEMs.Author and YearPatientsSampleMethodsRelevant MetabolitesBiological MeaningBridget M Stroup et al., 2018[29]10 patients with PKU and 15 controlsPlasma and urineGC/MS↑urinary TMAO in patient with AA-MFs compared with GMP-MFs↓deoxycarnitine in both diet in PKU patient↑Phenylacetate in PKU↓plasma total cholesterol in classical PKU compared to variant PKU and controls↑isoprenoid intermediate 3-hydroxy-3-methylglutarate in variant PKULower bioavailability of carnitine in AA-MFs diet compared with GMP-Mfs: despite higher l-carnitine supplementation in AA-MFs than in GMP-MFs, there’s a reduced bioavailability of carnitine from AA-MFs due, to degradation of carnitine to proatherosclerotic TMAO by intestinal bacteria.Reduced endogenous synthesis of carnitine in PKU patients compared with controlProve to support that the phenylacetate inhibits endogenous carnitine synthesisDifference between classical and variant PKU↑Endogenous cholesterol synthesis in variant PKUDenise M. Ney et al., 2018[31]27Plasma (18 patient) and urine (9 patient)GC/MS40 microbiome-associated metabolites in plasma:7 of them with different plasma levels in the pastiche with ingestion of AA-MF compared with GMP-MF:-↑3-methoxyltyrosine: in AA-MF compared with GMP-MF.-Plasma serotonin ↑ in variant PKU with ingestion of GMP-MF compared with AA-MF and with the ones with classical PKU.-↑Quinolinic acid in AA-MF compared with GMP-MF.45 microbiome-associated compounds in urine: 7 of these showed differential levels with AA-MF compared with GMP-MF:-↑tyramine and phenol sulfate were 50–90% higher with ingestion of AA-MF compared with GMP-MF.-↑kynurenine in AA-MF compared with GMP-MF.-↑Urinary excretion of quinolinic acid was significantly higher with AA-MF compared with GMP-MF.-↑6-sulfatoxymelatonin excretion: in female with classical↓6-sulfatoxymelatonin excretion: in male with classical PKUChanges in the intestinal microbiota with ingestion of AA-MFs diet result in degradation of Tyr and Trp reducing their bioavailability for neurotransmitters’ synthesisDifferent metabolic pathways in variant PKU compared to classical PKU.Difference between AA-MF and GMP-MFDifference in Tryptophan levels from serotonin synthesis in AA-MF compared with GMP-MF.Jeannette C. Bleeker et al., 2020[32]5 patients with VLCADDPlasmaUPLC-MScoupled to Thermo Q Exactive (Plus) Orbitrap mass spectrometer-↑β-ydroxybutyrate in KE (BHB-(R) 1,3-butanediol ketone ester) +CHO (carbohydrate)↓Intramuscular glycolytic intermediates (Hexose-P, Fructose-1,6-diphosphate, Glyceraldehyde-3P, 1,3-Diphosphoglyceric acid, 2-/3-Phosphoglyceric acid and Phosphoenolpyruvate) in KE + CHODifference in CHO and KE + CHO dietRaaschou-Pedersen DE et al., 2022[33]3 patients with PFKD and 3 controlsPlasmaGC–C–IRMS-↑C7Evaluation of the compliance to the treatmentMadsen et al., 2019[34]19 patients with McArdle disesease with 2 groups: 1 with triheptanoin treatment and 1 with placeboPlasmaGC-MS↑Malate and C5-ketones-↑AcylcarnitinesEvaluation of the compliance to the treatmentLokken et al., 2022[35]8 patients with GSDV and 4 controlsPlasma and breathGC-MS↑Pyruvate, lactate, and AcAc-HOB, glucose, and glycerolExercises changes and differences between the oral ketone ester supplementation and the healthy controlStorgaard J.H. et al., 2022[36]8 patients with fatty acid oxidation disorderPlasma and breathLC-MS/MS,GC-MS/MS,GC-IRMS↑Glucose, ↑glycerol,-=palmitateDifference between the RSV treatment group and the placebo one↑: increased value, ↓: decresed value, =: equal value.


In the last years, metabolomics studies have focused on understanding the efficacy, collateral effect and difference between different nutritional regimens.

Indeed 2017, a metabolomic study analyzed metabolites and neurotransmitters derived from Tyrosine and Tryptophan in plasma and urine samples from subjects with PKU consuming both AA-MF and GMP-MF. These authors confirm their hypothesis that differences in the composition of the intestinal microbiota occur with ingestion of AA-MF compared with GMP-MF, with a higher degradation of Tyr in AA-MF diet, thus reducing its bioavalability for synthesis of neurotransmitters [28]. Therefore, the prebiotic properties of GMP in conjunction with consumption of smaller amounts of Tyr distributed more frequently across the day appear to account for higher bioavailability of Tyr from GMP-MF [31].

Moreover, a year later, another study investigated adults and adolescents with PKU on a low-Phe diet and with AA-MF and GMP-MF to evaluate lipid metabolism though metabolomics analysis. They observed significantly difference in total plasma cholesterol concentrations in classical PKU, variant PKU patients and controls [29].

While, in patients with fatty acid oxidation (FAO) disorders, metabolomic was used to evaluate changes in metabolic pathways according to a different diet approach with the ingestion of ketone esters (KE). In fact, in patients with a maladaptive shift from fat to carbohydrate (CHO) oxidation during exercise this is thought to underlie myopathy and exercise-induced rhabdomyolysis. Thus, they tested this hypothesis in patients with very long-chain acyl-CoA dehydrogenase deficiency (VLCADD) [32]. The results described in this paper provide strong evidence in favor of KE supplementation before physical activity in patients with defective FAO at the level of VLCAD. The potential benefits of synthetic KE supplementation are probably not limited to FAO disorders such as VLCADD, but could be very relevant in other inborn errors of metabolism [32]. In these type of nutritional investigations, metabolomics can help to detect new metabolites or the variation of the known ones in the studied groups. For example, to study the effects of triheptanoin in Phosphorfructokinase deficiency and McArdle disease, targeted metabolomic was used to detect C7 ketones and CO2 in the first study (indicating the appropriate compliance to the diet), and the acylcarnitines in the second one, respectively. Both studies aimed to know if triheptanoin assumption could or could not improve the exercise tolerance [33,34].

Targeted metabolomic was also applied in another recent study on oral ketone ester supplementation in patient with McArdle disease, to detect pyruvate, lactate, and acetoacetate in plasma and hydroxybutyrate, glucose, and glycerol in plasma and in breath samples [35].

Finally, another application of metabolomics in this field, is represented by a study on resveratrol as a supply that can improve exercise capacity in patients with fatty acid oxidation (FAO) disorders [36].

## 4. Untargeted Metabolomic

In the less studied IEMs, the studies still focused on researching some diagnostic and severity markers. In this process, targeted and untargeted metabolomics has been widely used. In 2021, Ning Liu et al., to examine the utility of untargeted metabolomics, tried to use the traditional metabolomic screening approach for many potentially treatable IEMs not included on NBS panels, nor covered in standard, first-line biochemical testing. A total of 14 IEMs were detected, including 3 conditions not included in the Recommended Uniform Screening Panel for NBS. Screening also identified 128 unique cases with IEMs, giving an overall diagnostic rate of 7.1%. In total, 70 different metabolic conditions were determined, including 49 states not presently included on the Recommended Uniform Screening Panel for NBS. These findings suggested that untargeted metabolomics provided a 6-fold higher diagnostic yield than the conventional screening approach and identified a broader spectrum of IEMs. Notably, with the expansion of NBS programs, traditional metabolic testing approaches identify few disorders beyond those covered on the NBS. These data support the capability of clinical untargeted metabolomics in screening for IEMs and suggest that broader screening approaches should be considered in the initial evaluation for metabolic disorders (Table 4) [37].

For example, in IEMs of the non-oxidative branch of the pentose phosphate pathway, untargeted metabolomics was able to identify novel biomarkers, thereby expanding the current knowledge of TKT e TALDO conditions and providing further insight into potential underlying pathophysiological mechanisms. Furthermore, untargeted metabolomic testing offers the advantage of having a single effective biochemical screening test for identification of rare IEMs, that may otherwise go undiagnosed due to their generally non-specific clinical presentations [38].

Finally, the untargeted approach can also assess further perturbations of the metabolic network of disease in patients under ongoing treatment. For example, the metabolic defect in glycogen storage disease type I (GSDI) results in fasting hypo-glycemia and typical secondary metabolic abnormalities (eg, hypertriglyceridemia, hyperlactatemia, hyperuricemia). Thus, GSDI has profound effects on various metabolic pathways in addition to the known typical exceptions. These alterations are present despite optimized dietary treatment, which may contribute to the risk of developing long-term complications, an inherent problem of GSDI which appears to be only partly modified by current therapy [39].

## 5. Network Profiling Methods

Today the challenge is to interpret the metabolic pathways involved in each metabolic pathology. Indeed, metabolite perturbations are assessed based on current knowledge of specific metabolic deficits. Furthermore, the qualitative diagnostic process may have limitations that make it difficult to evaluate the clinical aspects observed in the patient.

To address this gap, Karlien et al. proposed a single-platform, non-targeted quadrupole time-of-flight (QTOF) LC metabolic profiling method that can be applied for diagnostic screening of metabolic deficits present in each patient called Next Generation Metabolic Screening (NGMS).

Through disease-specific filters, the authors identified a panel of 340 clinically relevant metabolites known to be diagnostic or to evaluate the efficacy of therapeutic interventions. The method would reduce the complexity of the data obtained and to extract the necessary diagnostic information.

Thus, targeted analysis can be used to confirm the specificity of new disease biomarkers. For example, plasma NGMS analysis or evaluating of metabolic pathways of genetic variants of uncertain significance are applicable to type II xanthinuria. NGMS can also be used for non-targeted metabolomic results, be filtered using a panel of disease-associated metabolites. In fact, with this approach it was possible to obtain a correct diagnosis for 42 out of 46 IEMs by identifying potential new biomarkers known and still unknown [15].

Another example concerned the identification of N-acetylmannosamine, which was increased in the body fluids of patients with severe developmental delay and dysmorphism. This marker identified a new metabolic pathology due to N-acetylneuraminic acid phosphate synthase (NANS) deficiency (Table 5) [15,40].

Such metabolites relationships can also be formalized as networks, where the nodes correspond to the metabolites or features (when there is no or only partial identification), and edges connect nodes if the corresponding metabolites are related. Several networks can be built from a single dataset (or a list of metabolites), where each network represents different relationships, such as statistical (correlated metabolites), biochemical (known or putative substrates and products of reactions), or chemical (structural similarities, ontological relations) [41].

Moreover, in 2022, Thistlethwaite et al., tried to improve upon manual diagnosis of IEMs in the clinic by developing novel computational methods to analyse untargeted metabolomics data. They employed an automated computational diagnostic method that “connects the dots” (CTD) between metabolite perturbations observed in individual metabolomics profiling data and modules identified in disease-specific metabolite co-perturbation networks learned from prior profiling data. Their approach can be used to supplement information from biochemical pathways and has the potential to significantly enhance the interpretation of variants of uncertain significance uncovered by exome sequencing [42].
metabolites-13-00447-t005_Table 5Table 5Relevant Network applied methods studies applied to the IEMs investigation.Author and YearPatientsSampleMethodsRelevant MetabolitesBiological MeaningKarlien L. M. Coene et al. 2018[15]92: (46 patients with different IEMs 46 controls)PlasmaLC-QTOF↑N-methylnicotinamide, imidazole,Lactic acid, N-acetylmannosamine↓N-methylpyridone, carboxamideshydantoin, propionic acidN1-methyl-8-oxoguanine,pyrrolidine-2-one,pyridoxatein different IEMsMarkers of a specific IEM or involved pathways.Lillian R. Thistlethwaite et al., 2022[42]539 samples:From meta-analysis of untargeted metabolomics studies and unreported samplesPlasmaCTD MmethodCorrect diagnosis in 79% of the remaining 137 samples with known diagnoses across the remaining 15 modeled IEMs,94% of samples hadthe correct diagnosis-in their differential diagnosisshort list.Powerful when applied to individuals who are undiagnosed by current methods↑: increased value, ↓: decresed value.


The benefit of CTD-based metrics can be compelling when applied undiagnosed individuals by current methods. By quantifying the likeness of individuals metabolite perturbations with perturbation patterns observed in many diseases, candidate diagnoses can be ranked and possible diagnoses can be recommended. Furthermore, if genetic sequencing data are available for individuals exhibiting strong disease-specific metabolite perturbation patterns, VUSs can be re-interpreted given the available evidence provided by untargeted metabolomics. While CTD-based metrics cannot eliminate manual review entirely, they can expedite it and increase the confidence by which clinical laboratory directors make diagnostic decisions. Finally, disease-specific network models can be automatically and continuously updated as new case profiling data accumulates, ensuring stronger network stability and improved diagnostic performance [42].

## 6. Future Perspectives

The expansion of metabolomic in clinical biochemistry and laboratory medicine communities as brought these approaches into clinical practice. Metabolomic has a part in newborn screenings in most developed countries and an important role in the differential diagnosis between IEMs. Moreover, Metabolomic can help in evaluating metabolites in follow-up as markers of severity or efficacy of diet and therapies. It is still crucial detecting new IEMs while performing untargeted metabolomic studies, new markers of well-known IEMs and in the characterization of metabolic pathway diseases to find new or different therapy approaches. Lastly, several research groups are trying to profile metabolomics, proteomics and genomics data in network platforms to have a holistic vision of of every single patient.

In conclusion, metabolomics has a predominant and well-characterized role in IEMs, both in scientific research and current clinical practice.

## Figures and Tables

**Table 1 metabolites-13-00447-t001:** Relevant targeted metabolomics studies with cerebrospinal fluid samples.

Author and Year	Patients	Sample	Methods	Relevant Metabolites	Biological Meaning
Glynis Klinke et al., 2020 [20]	131 samples	Cerebrospinal fluid	LC-MS/MS	38 metabolites from Aromatic l-amino acid, decarboxylase deficiency, guanidinoacetate methyltransferase deficiency, ornithine aminotransferase deficiency, cerebral folate deficiency and methylenetetrahydrofolate reductase deficiency.	analytical validation, establishment of literature-based CSF cut-off values and reference ranges, of available CSF samples obtained
Hanneke A Haijes et al., 2019 [21]	41 (30 controls, 11 patients)	Cerebrospinal fluid	DI-HRMS	Non-ketotic hyperglycinaemia: ↑Glycine Propionic aciduria: ↑Propionylcarnitine, ↑2-Methylcitric acid, ↑3-Hydroxypropionic acid, ↑Propionylglycine, ↑Heptadecanoylcarnitine, ↑Glycine, ↑Propionic acid, ↓l-Carnitine Purine nucleoside phosphorylase deficiency: ↑Inosine, ↑Guanosine, ↑Deoxyinosine, ↑Deoxyguanosine, ↓Guanine, ↓Hypoxanthine, ↓Uric acid Argininosuccinic aciduria: ↑Argininosuccinic acid, ↑Citrulline, ↑Orotidine, ↑Glutamine, ↑Uracil, ↑Orotic acid, ↓Arginine Tyrosinaemia (type not specified): ↑L-Tyrosine, ↑Hydroxyphenyllactic acid, ↑Hydroxyphenylacetic acid - Type I (MIM #276700): ↑4-Hydroxyphenylpyruvic acid; - Type II (MIM #276600): ↓Decrease, product (II) - Type III (MIM #276710) ↑5-Aminolevulinic acid, ↑4-Fumarylacetoacetic acid, ↑Succinylacetone, ↓Fumaric acid, ↓Acetoacetic acid Hyperphenylalaninaemia, non-tetrahydrobiopterin deficient (genetic defect not specified): ↑Phenylalanine/Tyrosine ratio, ↑L-Phenylalanine, ↑Phenylpyruvic acid, ↑Phenylacetic acid - PAH: ↑Phenyllactic acid - DNAJC12: ↓Tyrosine Hypermethioninaemia: ↑Methionine, ↑Homocysteine, ↑Homocysteine thiolactone, ↓S-Adenosylmethionine.	Accurate biochemical profile in a set of patients in CSF, supporting the use of CSF metabolomics in metabolic diagnostic laboratory

↑: increased value, ↓: decresed value.

**Table 4 metabolites-13-00447-t004:** Relevant untargeted metabolomics studies applied to the IEMs investigation.

Author and Year	Patients	Sample	Methods	Relevant Metabolites	Biological Meaning
Ning Liu et al., 2021 [37]	4464 clinical samples - 70 c	Plasma and urine	GC-MS, LC-MS/MS	70 conditions: - 21 included and 49 not included in the RUSP, - 26 of these 49 are not covered - by traditional screening, - and 7 of these 26 disorders are currently treatable.	Untargeted metabolomics provided a 6-fold higher diagnostic yield compared with the conventional screening approach and identified a broader spectrum of IEMs.
Brian J. Shayota et al., 2020 [38]	5 patients with TALDO deficiency and TKT deficiency	Plasma and urine	LC-MS	↑arabitol/xylitol, ribitol, erythritol, ribose, ribonate and erythronate, kynurenine, xanthurenate, quinolinate, indolelactate, xanthosine, kynurenate, xanthurenate alpha-ketoglutarate and sedoheptulose in plasma ↑ribulose/xylulose kynurenine, 3-hydroxykynurenine, guanosine, quinolinate xanthurenate, - succinate, fumarate, malate and sedoheptulose in urine	Untargeted metabolomic approach useful in differential diagnosis for IEMs of the Pentose phosphate pathway
Tamara Mathis et al., 2022 [39]	45 samples: 14 patients and 31 controls	Plasma	UHPLC HR-MS/MS GC/EI-MS	Several altered metabolic pathways: Energy, Urea cycle, C1, amino acid Purines and pyrimidines metabolisms	Pathophysiology of glycogen storage disease type I: metabolites alterations are present despite optimized dietary treatment, which may contribute to the risk of developing long-term complications

↑: increased value.

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
