# Peer review of "Metabolomic Studies in Inborn Errors of Metabolism: Last Years and Future Perspectives"

_metabolites, 2023, doi:10.3390/metabo13030447_

Round 1

Reviewer 1 Report

Please see the attached document where there are marked inserts in the pdf manuscript for several recommendations or corrections.

Author Response

Thank you for your kind and relevant suggestions. We have made the necessary modifications to the text based on your comments. Specifically, we have compared the advantages and disadvantages of various techniques in lines 116-133, as requested. We have also rephrased the specific section you indicated, added missing references, and updated the author contributions as you recommended.

Reviewer 2 Report

You state that this is a review of studies available from 2018-2022 however several of you references were from earlier than this time period. These references would be in your earlier review paper so please go through these and update/correct.

I found the paper a little repetitive. You define that IEM are due to changes in metabolites in line 40, 59 onwards, and you define metabolomics in line 45, 80, 85, 95, 99. These are all saying essentially the same thing. You also then define targeted and untargeted metabolomics. This information is in your earlier paper so a lot of it can be referenced to that paper.

I also feel this is not a review but rather a list of the papers available on this subject. I wish to read about these studies and the impacts and advances they have made to the field rather than look at a long table without really getting a feel for the research. 

One of the points that seemed to be coming across was that you were looking at untargeted approaches which seemed to be where the research was headed this could be a good angle to give the review some purpose and structure.

There is obviously a large amount of work goes into a review but just feel there was no real story to the review to really see where the research is heading in this feel. I think it just needs a re-write with a little more thought to the overall picture.

Author Response

We would like to express our gratitude to the reviewer for your helpful comments. We have made the necessary modifications to the paper based on your suggestions. Additionally, we have attempted to present the currently available studies in a more natural and fluid manner. We have corrected the time period that the review focuses on as we realized that three articles published in 2017 were not included in our preview review of this topic.

We have also revised the definitions of IEMs and metabolomics to avoid repetition, as the reviewer's suggestion. The paragraph on targeted and untargeted metabolomics has been removed (lines 138-146) as it is cited in the chapter below (168-171 and 182-190). We have modified the text to focus on the meaning of the research and cut down on excessive details in order to provide a more comprehensive overview. Additionally, we have synthesized the table for clarity.

We acknowledge that these changes have made the text more readable and organic. Thank you again for your valuable feedback.

Reviewer 3 Report

The work provides a comprehensive view of the current state of metabolonics and also the perspective of its further use in theoretical research and in clinical practice. Regarding the scope and spectrum of its scope, metabolonics can be compared to genetics or endocrinology in the entire period of their first knowledge and current development. It is obvious that metabolonics has opened a new window to the knowledge of life, and the knowledge presented by the authors is very broad. However, it is obvious that the issue is in the stage of entropy and only further studies of top workplaces will make a synthesis of knowledge and its use in practice. From this aspect, the work is beneficial, but in many respects it goes into details that go beyond the scope of an overview paper. I propose to reduce the work - mainly the scope of Table 1.  

Author Response

Thank you for your kind comment. We have made the necessary modifications to the paper to provide a more comprehensive view and reduce the level of detail in every single article cited, as suggested (see lines 226-230, 250-255, and other chapters). Additionally, we have followed your advice and reduced Table 1. We believe that these changes have made the article more readable and effective as a review. Once again, thank you for your valuable feedback.

Round 2

Reviewer 2 Report

A much improved manuscript. However the table is very reader unfriendly. It is very difficult to follow particularly as the studies are in chronological order. It would make more sense and be much clearer if you order them in another way. Specifically by matching to your following headings 3.1, 3.2 and 3.3 and having 3 smaller tables would be a huge improvement for the reader to be able to glean information. Even ordering by sample type or method would be an improvement. The table spanning 8 pages is far too long.

The final section Future Perspectives needs to be corrected. There are quite a few typos

Line 397 should be has rather than as

Line 398 should be example rather than exam.

The first sentence is too long to convey meaning (line 396-400) Please consider breaking into 2 sentences.

Line 401 studies instead of study

Line 402 no s on pathways.

Line 403 What is the platform? Please clarify in the text.

Line 405 the metabolomic ??? (missing word here)

Line 406-407 this sentence is not needed. If it already has a predominant and well-characterised role then it is not one of the most promising approaches nor can it become the new chemistry.

Author Response

Dear Reviewer,

Thank you so much for your comment. We have revised the table according to the presentation order of the studies in the article. Additionally, we have subdivided the table into smaller ones, with one for each subchapter. I believe this approach will make it easier for readers to glean information while reading the article.

Furthermore, we have made the suggested corrections in the final section of the paper.

Thank you very much again for your support.